# Collaborative Fleet Deployment and Routing for Sustainable Transport

**Panagiotis Ypsilantis [†] and Rob Zuidwijk *,[†]** 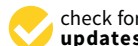

Rotterdam School of Management, Erasmus University, 3062 PA Rotterdam, The Netherlands;
pypsilan@gmail.com
* Correspondence: rzuidwijk@rsm.nl; Tel.: +31-10-408-2235
† The first author wrote his PhD thesis on the topic; the second author acted as supervisor.

**Abstract:** Efficient multi-modal transportation in the hinterland of seaport terminals depends on consolidation of container volumes in support of frequent services of high capacity means of transport, such that sustainable multi-modal transport can compete with uni-modal road transport in cost and time. The tactical design of barge scheduled transport services involves fleet selection and routing through the inland waterway network. The resulting network service design should meet expected demand and service time requirements set by the shippers. We develop a tight MILP formulation for the Fleet Size and Mix Vehicle Routing (FSMVRP) especially adapted for the Port-Hinterland multi-modal barge network design. Also, an analytical model is developed to help understand important design trade-offs made. We consider the case of horizontal cooperation of dry port container terminals that share capacity. Our results show that in case of cooperation, both cost savings and service levels are improved, and allow for sustainable multi-modal transport to be competitive with uni-modal truck transport.

**Keywords:** sustainable transport; multi-modal transport; port-hinterland logistics; mixed integer linear program

---

## 1. Introduction

The containerization of cargo has enabled a vast increase in international maritime trade whilst reducing the associated transportation costs [1]. However, the first and the last leg of the international door to door maritime container transport, i.e., transport between sea ports and locations inland, have not witnessed similar efficiency gains. This despite contributing significantly to the total transportation cost and total lead time, even though the distances covered there are relatively small [2].

The performance of these port-hinterland transport legs depends on the deployment of transport resources and the design of transport services on the inland network to meet demand. The inland transport may involve multiple means of transportation. River vessels (barges) and trains transport containers between sea ports and inland terminals, and trucks between inland terminals and distribution centers or other locations where goods are received or shipped. Transportation by truck only is also feasible, but is both more costly and associated with other negative externalities such as emissions and road congestion [3].

It is generally believed that inland waterway and rail transport are more sustainable than road transport [4]. Although emissions per ton-kilometer are currently lower for barges and trains than for trucks, such figures must be considered with care [5]. For instance, actual distances traveled and use of the barges need to be taken in consideration as well. Moreover, the advancement of vehicle technologies and regulations [6] may help reduce the emission factors of trucks considerably.

In any case, to foster the alternatives to road transport, it is important that these multi-modal transport services are competitive. Shippers will require multi-modal transport to be just as efficient and reliable as truck transport, but will then favor more sustainable options. The shift of container transport from road to inland waterway and rail was promoted by the Rotterdam Port Authority. They imposed that at least 20% of import and export container volume through the port should be transported by rail, and at least 45% by inland waterways by 2035. In 2016, 10% was transported by rail and 36% was transported by inland waterways.

Inland terminals that connect to a sea port through a high capacity transport corridor and as such transfer the port hub function inland, are referred to as "Dry Ports" [7,8] or "Extended Gates" [9]. While the core activity of container terminals is the handling and storage of containers, dry port operators design transport services on the inland network to meet demand, supported by a fleet of barges, trains and trucks.

The establishment of such a transport corridor involves collaboration among multiple organizations to synchronize various transportation and transshipment activities. As a result, the design of multi-modal services is quite complex since the perspectives of several actors need to be taken into account [10].

In this paper, we will focus on barge transport, but results are applicable to rail transport as well. The effective deployment of barges depends on the consolidation of container flows such that economies of scale are achieved while providing frequent services to customers. Business examples, such as the development of multi-modal transport services by Hutchinson Ports in the port of Rotterdam [11] show that when implemented successfully, multi-modal transport can compete with uni-modal truck transport both in cost and time even for relatively short distances.

Consolidation of flows is usually achieved in two ways. First, hub and spoke networks can be developed that achieve economies of scale by concentrating flows in corridors that connect the main hubs. Second, the rotation of barges along several terminals provides aggregate capacity to serve transport demand between various origins and destinations.

The customers of multi-modal transport services, the shippers, opt for low costs and low transit times. Multi-modal transport services should therefore not only achieve economies of scale to reduce costs, but also provide high frequency of service, as time to next departure adds to the transit time. Ehrfurt and Bendul [12] demonstrate that higher frequency of service also improves reliability of transportation services.

Therefore, the main question to be addressed in barge network service design is: *How to balance the various network service design possibilities such that barges are deployed at minimum costs and such that service requirements are met?* Konings [10] developed a framework that included such network design possibilities. Inspired by this framework, we summarize the design parameters and performance indicators in barge network design in Table 1. The economies of scale emerge from the deployment of bigger barges that are well-used. Barge size or capacity is measured in standard container volume units, Twenty feet Equivalent Unit (TEU). Barge sizes in our case study analysis range between 30–150 TEU. Not all containers transported are equal to an integer multiple of the standard loading unit TEU, and the amount of cargo carried by a transport means may be constrained by weight instead of volume. Nonetheless, in this paper, as is often done in practice and academic literature, we express both demand volumes and capacities (of barges) in TEU. The use of barges depends on the fill rates, but also on the number of round-trips per planning horizon that can be achieved given the circulation times of the round-trips. When high demand flows are concentrated in a particular origin destination pair, point-to-point services can be formed that keep circulation times low. However, when demand flows are relatively low, barges may need to rotate along several terminals in order to achieve high fill rates. Currently, a barge visiting the port of Rotterdam, for instance, calls on average eight terminals [13]. This results in higher circulation times, as barges need to berth at these terminals. The circulation time increases in particular due to waiting times and delays at seaport terminals, for example caused by deep sea vessels claiming priority over barges. To address waiting times in the port, Konings [14] proposes the reorganization of services of barge operators by splitting services in trunk line operation

in the hinterland and collection/distribution services in the seaport area. In such a way, fewer calls are performed in the seaport area so the circulation time of barges decreases.

**Table 1.** Barge design parameters and performance indicators.

| Design Parameters | Performance Indicators |
| --- | --- |
| Number of Barges | Costs |
| Size of Barges | Capacity Installed |
| Number of calls | Network Coverage |
| Distribution of calls between inland and seaport terminals | |

The performance indicators in Table 1 are explained here. Costs for operating barges and trucks include costs of assets, crew and fuel, with a markup for the transport operator. The Capacity Installed on the network refers to the sum of the capacities of the barges, each multiplied by the number of times the corresponding barge is deployed on a route within the planning horizon. The Network Coverage equals the average frequency of service offered to OD pairs in the network.

This paper will take the perspective of the dry port operator who aims to design its barge services on the inland network in an optimal way, while considering costs and shipper preferences. Negative externalities are not minimized by the dry port operator, but can be analysed afterwards as collateral benefits. Port-hinterland container transport demand can be satisfied by either combined barge-truck services or by direct trucking, since shippers are usually indifferent to the mode used as long as cost and service requirements are met.

In our paper, we aim to demonstrate that cooperation among dry ports is beneficial, and we aim to identify the trade-offs that drive optimal inter-modal network design. The formulation of an optimization model will help us to identify the benefits of cooperation by comparing the outcomes of cooperative dry ports with the case of independent dry ports. We develop an analytical model, which roughly approximates the optimization model, to help us understand the trade-offs that drive optimal network designs. We now discuss the two modeling efforts in more detail.

First, we formulate a Mixed Integer Linear Program (MILP) model for the Fleet Size and Mix Vehicle Routing problem (FSMVRP) with multiple depots, adapted to the design of port hinterland multi-modal network services. The model aims to support decisions regarding: (1) the fleet size and mix selection, (2) routing of the fleet over the network within a planning horizon in order to satisfy demand under service frequency constraints, and (3) the assignment of container flows to these services in order to assess the performance of the proposed network design. FSMVRP models are NP-hard since they can be reduced to the VRP. We take advantage of the special structure of the problem in our case by the use of some artificial variables. We provide a compact MILP formulation that enables the easy construction of round-trips that can be solved with commercial solvers. Moreover, the model is applied to a real case of an alliance of clustered dry ports located in the Brabant region of The Netherlands that connect with container terminals located within the port of Rotterdam. We investigate the impact of cooperation, i.e., sharing transport capacity among these dry port terminals, on cost and service frequency. We consider the construction of round-trips in depth by investigating their relationship with other variables, such as the size of barges deployed and the expected service times. Our results show that cooperation between clustered dry ports enhances their performance, especially when higher minimum frequencies are imposed. Moreover, we observe that in scenarios with severe delays at the seaport terminals there is a shift of the location of calls from seaport terminals to inland terminals. Thus, a lower number of calls at the seaport terminals are realized resulting in more barge trips.

The aforementioned optimization model provides optimal barge fleet deployment and services on the network. However, we aim to analyze the structure of these optimal solutions in terms of trade-offs, and we will do so by means of an analytical model in Section 5. The analytical model can be considered a stylized version of the optimization model introduced in Section 4. The analytical

model, due to its simple structure, helps us to better understand the trade-offs made in network design. Although the analytical model provides outcomes similar to those of the optimization model qualitatively , the analytical model does not provide optimal solutions. The optimization model is of value in that respect.

There are several trade-offs that need to be made in the deployment of a barge fleet and in the design of barge services on a network. Fleet selection and routing decisions are interrelated, as they emerge from optimal solutions to the MILP presented in Section 4. As shown analytically in Section 5, these design variables are the main determinants of all cost, capacity and service time performance indicators. The basic design trade-offs are:

- Fleet composition: the use of big barges to reap economies of scale and of small barges to operate at a higher frequency;
- Fleet routing: routes with many stops to consolidate demand and to provide high frequency of services versus routes with few stops to have short circulation times.

Both trade-offs need not be made similarly for all barges and all routes, so that the fleet composition and its routing are a mixture of these trade-offs.

Section 2 discusses literature on relevant models to the port hinterland network design. Section 3 discusses academic and managerial impacts of the paper. The MILP model formulation will be explained in Section 4. In Section 5, results are discussed. In Section 6, we develop the aforementioned analytical model. Finally, Section 7 provides the conclusions.

## 2. Literature Review

The development of the supply side of container transport networks was studied extensively in the literature and is widely known as the service network design problem. Such problem formulations are increasingly used to address the tactical issues of carriers according to Crainic [15]. Recent overviews of multi-modal freight planning research are conducted by Caris et al. [16] and SteadieSeifi et al. [17], while Crainic et al. [18] develops a taxonomy for inter-modal freight transport simulation models. The authors divide the contributions in the field according to the time horizon in strategic, tactical and operational models. Strategic decisions in multi-modal transportation usually relate to long term decisions such as node and link infrastructure investments. Operational decisions in this context come down to assigning containers to specific transport itineraries such that capacity is effectively used and time constraints set by the shippers are met.

Tactical barge network service design includes the optimal selection and routing a fleet, and the offering of transport services executed by the fleet against lowest possible cost and in compliance with service level requirements. The general case of such problems is addressed in the literature by the solution of the Fleet Size and Mix Vehicle Routing problem (FSMVRP). The FSMVRP was introduced by Golden et al. [19] as an extension of the vehicle routing problem by considering a heterogeneous fleet of vehicles. Salhi and Rand [20] review models on vehicle fleet composition problems and highlight the importance of incorporating vehicle routing in such decisions. Since its introduction, several extensions of the problem have been proposed in the literature, such as considering multiple depots by Salhi and Sari [21] or the consideration of time windows by Liu and Shen [22]. Given the difficulty of solving such problems, literature contributions in this field are mainly focusing on the development of efficient heuristic procedures for the solution of the problem.

Crainic and Kim [23] identify the consideration of time in multi-modal transport models as a major research challenge. The use of time in such models is twofold. First the scheduling, coordination and routing of transport modalities should also be assessed over time. Second, transit times faced by the customers of the transport services should be considered since different modalities and different multi-modal paths result in different transport, dwell and delay times.

Contributions that study port-hinterland multi-modal network service design are still limited. Relevant literature includes models focusing on corridor design, line bundling, and the design of

hub and spoke networks. Relevant contributions focus on two main elements crucial to the effective representation of multi-modal systems. First, the characteristics of different modalities should be effectively formulated and their use should be assessed both in terms of cost and time. Second, the demand penetration of multi-modal services compared to that of uni-modal truck transport should also be assessed both in terms of costs and of service levels from the customer's perspective.

In Table 2, the main research done on Port-Hinterland network design is summarized, and each paper is briefly discussed. The models considered differ in several dimensions including the planning level, the mathematical formulation, the solution approach, and other modeling considerations.

We found several contributions that consider the barge network design problem at the operational level. Crainic et al. [24] discuss the optimization challenges that arise by the development of the dry port concept and proposes a service network design model, in a space-time network, for the operational rotation planning of barges between seaport and inland terminals. The size of the problem becomes restrictive even for small and medium instances due to its space-time format, so commercial solvers fail to find feasible solutions. Further research on the development of efficient solution methodologies for such problems in space-time formulations is needed.

**Table 2.** Comparison of models in Port-hinterland network design.

| Paper | Objective | Formulation | Level | Means of Transport | Modeling Considerations | Time | Solution |
|---|---|---|---|---|---|---|---|
| Crainic et al. [24] | Minimize operational cost | Service Network design for barge transport | Operational | Barge Truck | Space time network is considered | Space time network is considered <br> Time windows | Heuristic procedures |
| Sharypova et al. [25] | Minimize operational cost | Continuous time formulation of the service network design problem | Operational | Barge | Routes are constructed Transshipments are allowed <br> One round-trip per vehicle used | Synchronization and Coordination of barges <br> Time windows for the pick up and delivery of commodities | Heuristic procedures |
| Fazi et al. [26] | Minimization operational costs | Heterogeneous Vehicle Routing problem | Operational | Barge | Allocation of containers to heterogeneous barge fleet | Comparison of various planning horizons with release and due dates for the containers | Heuristic procedures |
| Heggen et al. [27] | Minimize total transport costs | Integrated Inter-modal Container Routing | Operational | Train Truck | Comparison of sequential and integrated planning of long-haul rail and truck drayage operations | Time windows | Heuristic procedure |
| Caris et al. [28] | Corridor Network Design Concave cost function (Economies scale) | Path-based multicommodity network design formulation | Tactical | Barge | round-trips and base costs given <br> Not Elastic demand | Not considered | Commercial Solver |
| Braekers et al. [29] | Optimal shipping routes and barge size | Network design model adapted for line bundling | Tactical | Barge | round-trips per week given <br> One type of barge <br> No service quality measure | Time per round-trip is considered and bounded from above | Commercial Solver |
| Ypsilantis and Zuidwijk [30] | Joint capacity setting and pricing of corridors | Bi-level MIP <br> 3 decision actors Operator, Competition, Shippers | Tactical | Barge Train Truck | Shuttle services, No Rotation <br> Tradeoff: Economies of Scale, Service level | Frequency dependent expected dwell times <br> Maximum number of round-trips given | Heuristic procedures |
| Riessen et al. [31] | Capacity setting on corridors | Path based network design MIP model | Tactical, Operational | Barge Train Truck | Self Operated and Sub-contracted services | Service time of paths is given Flexible due dates, Penalty costs | Commercial Solver |
| Wang and Meng [32] | Minimize network construction and operational costs | non-linear MIP model | Strategic, Tactical | Truck Rail Sea | Route choice behavior of intermodal operators | Transit times are taken into account | Heuristic procedures |
| This paper | Joint Fleet Composition and Routing | Continuous time MILP | Tactical | Barge Truck | Construction several barge round-trips per planning horizon | Utilization of barges in both space and time levels | Commercial Solver |

Sharypova et al. [25] develop a continuous time formulation for the scheduled barge network design problem with synchronization and transshipment constraints. This model can facilitate the operational planing of barge routing. The heterogeneous fleet of barges is routed through the network and the arrival and departure of each barge at each node are specified under a big set of synchronization and coordination constraints. Demand, organized in commodities, is assigned to transport services that satisfy pick up and delivery time windows. The size of the problem allows the treatment of only very small instances with commercial solvers. Therefore, Sharypova [33] develops some meta-heuristic approaches. Problems at the operational level do not allow for simplifications that reduce the computational complexity, so heuristic procedures are needed for the solution procedure.

Fazi et al. [26] studies the allocation of containers to a given heterogeneous fleet of barges along several given routes. Consequently, the paper focuses on the operational planning problem.

Heggen et al. [27] proposes an inter-modal routing problem where long-haul inter-modal transport and drayage routing are integrated. Sequential planning, i.e., first long-haul rail transport and then truck drayage, is compared with integrated planning. The advantages include better use of rail sets and possibly more efficient routing of trucks.

Most of the models discussed in the literature address problems at the tactical level. At this level, the formulation of economies of scale to better capture the expected cost of solutions, the characteristics of different modalities, and the competition between combined transport and uni-modal trucking become crucial.

Caris et al. [28] adapt the generic path-based multicommodity network design formulation of Crainic [15] to multi-modal barge transport by using a concave cost function to formulate economies of scale for the links operated by barge. The element of time is ignored in this formulation. The impact of cooperation of barge inland terminals is assessed only in terms of consolidating flows on some corridors, but an analysis of overall transport performance and costs or routing are not considered.

Braekers et al. [29] develop a line bundling MIP model to construct round-trips of barges and assign container flows to round-trips in a tactical planning horizon. The model inputs include the weekly number of trips, and thus the maximum round-trip time, and the number and size of barges. Customer demand should always be fulfilled by one round-trip and trucking is not considered to be a recourse action. The authors run several scenarios to assess the optimal capacity setting on the corridor. Their results are in favor of larger barges to leverage economies of scale. However, barge operators may in practice favor using smaller barges to guarantee higher frequency services to be more attractive to customers, as noted previously.

Ypsilantis and Zuidwijk [30] develop a model for the joint design and pricing of multi-modal shuttle barge services according to the extended gate concept. The bi-level structure of their model allows on the first level the capacity setting on corridors by selecting the size of modalities and frequencies of connections, in parallel with the pricing of the services offered to customers. On the second level, customers select the minimum cost paths that satisfy their service time constraints among ones operated by the extended gate operator, but also by other transport providers, such as trucking companies. Both costs and expected service times on corridors are dependent on both frequency and means of transport. Penetration of combined transport services is therefore achieved both in terms of costs and time.

Riessen et al. [31] propose a path-based service network design model that investigates the use of contracted and subcontracted network services for the operation of an extended gate network at a tactical level, while assuming flexible due dates. Their findings show that transshipment cost at terminals should be reduced in order for paths with more than one stop at inland terminals to become cost effective.

Wang and Meng [32] propose an non-linear MIP model formulation for a intermodal network design problem, which incorporates economies of scale, congestion effects, and random route choice behavior. The model does not consider the rotation of vehicles to provide transport capacity on the links.

Summarizing the literature review, some conclusions can be drawn. Although the supply side of transport networks have been extensively studied in the literature, the design of port-hinterland multi-modal transport systems has only recently grasped the attention of the academic world. The models proposed are most of the time between the tactical and the operational levels. Although they aim to support tactical decisions, such as capacity setting or fleet composition, specific demand instances are considered and the models are solved based on minimizing operational cost and meeting operational time constraints. Moreover, most models seem to ignore crucial factors to the multi-modal nature of the problem such as the consideration of time constraints, or are too computationally intensive when time is taken into account.

## 3. Academic and Managerial Impacts

Our optimization model outputs concern the tactical joint fleet deployment and routing of barges in port hinterland networks. The optimal fleet should be selected based on its expected operational performance and that is why routing plans for barges should be considered simultaneously. In this sense, the use of the fleet should not only be based on the consolidation of flows but also on the construction of round trips that achieve low circulation times and thus more round trips per barge within the planning horizon. Moreover, by considering the routing of the fleet, we look at the expected service level that shippers of the network will experience by controlling the minimum frequency per OD pair serviced.

As direct trucking is always an alternative to multi-modal transport, the modal split between barge and truck transport follows from cost and service time performance of these alternatives. The optimal barge network design seems to depend, on the demand consolidation opportunities, on the characteristic of the waterway network (distances, depth, bridges, locks, etc.), and on demand characteristics, such as service time requirements, cost elasticity, service time elasticity on different modalities. Our research contributes to existing literature by proposing models that take all the above factors into consideration.

The paper also contributes by the identification of benefits of cooperation among dry ports by comparing the model outcomes of cooperative dry ports and independent dry ports, using the aforementioned optimization model.

The paper finally provides a detailed analysis, using both the optimization model and an analytical model, of the underlying trade-offs that drive optimal inter-modal network designs.

The identification of basic trade-offs will help managers to intuitively understand the rationale behind optimal network designs. They may also use the optimization model outcomes as an input to their design considerations, although this would require the model to be validated for the specific business context at hand.

## 4. Mixed Integer Linear Program Model

We develop a model for the joint barge fleet selection and routing in order to provide an optimal scheduled service network design. The use of the fleet is not only driven by the fill rates of the vessels, but also by the time use based on the number of trips that can be scheduled within the one-week planning horizon.

As discussed already in the literature review section, the FSMVRP is NP-hard and usually difficult to solve even for small and medium instances. To solve our problem, we formulate a compact and tight MILP such that it can be solved to near optimality with commercial MILP solvers. We achieve this by three main modeling techniques. First, we exploit the spatial structure of our network to reduce the number of viable round-trips for barges. This can be achieved by the introduction of some artificial nodes through which all barges routes need to pass. Second, we reduce the number of commodities by aggregating them as the expected weekly demand of all customers in Origin-Destination (OD) pairs that require a given number of services over the planning horizon. Third, we provide an upper bound for our objective given by the costs of uni-modal truck transport for all OD pairs.

## 4.1. Notation

The construction of the barge routes is presented schematically in Figure 1. During the round-trip, the barge loads export containers at the inland terminals and discharges them at the seaport terminals, where it loads import containers that are discharged at the inland terminals. Some inland terminals can be visited twice on a given route, while each seaport terminal can be visited only once.

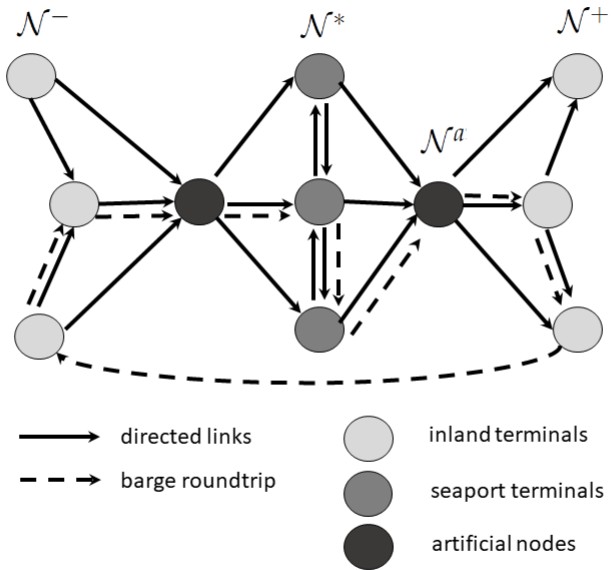

**Figure 1.** Construction of barge routes

We now elaborate on model notation of sets, cost and capacity parameters, time parameters, and decision variables. Tables 3–7 summarize the notation used in the model.

### 4.1.1. Sets

Let us consider an underlying directed network $\mathcal{G} = (\mathcal{N}, \mathcal{A})$ with node set $\mathcal{N}$ and arc set $\mathcal{A}$.

**Table 3.** Sets.

| | |
|---|---|
| $i \in \mathcal{N}$ | Nodes |
| $i \in \mathcal{N}^- \subseteq \mathcal{N}$ | Nodes representing inland terminals at the start of the round-trip |
| $i \in \mathcal{N}^+ \subseteq \mathcal{N}$ | Duplicate nodes representing inland terminals at the end of the round trip |
| $i \in \mathcal{N}^* \subseteq \mathcal{N}$ | Nodes representing seaport terminals |
| $i \in \mathcal{N}^a \subseteq \mathcal{N}$ | Nodes representing artificial nodes that are used to connect the inland terminal region to the seaport terminal region |
| $(i, j) \in \mathcal{A}$ | Arcs on the network |
| $c \in \mathcal{C}$ | Commodities |
| $b \in \mathcal{B}$ | Barges |
| $r \in \mathcal{R} \subset \mathbf{Z}_{>0}$ | Barge round-trips |

### 4.1.2. Commodity Parameters

We consider the multi-commodity formulation of the problem in which each commodity, $c \in \mathcal{C}$, is associated with the expected weekly container demand for a specific Origin and Destination (OD) pair, $(O^c, D^c) \in \mathcal{N} \times \mathcal{N}$, under some service time constraints. The demand volume of such a commodity $c$ expressed in TEU is denoted by $d^c$, and represents the level of demand for both inbound and outbound flows regardless of whether the containers are full or empty. The import and export flows of containers are assumed to be balanced, since any import flow of full containers would lead to the return of empty containers and vice versa. In reality, some empty containers dwell at the inland

terminals until some demand for export containers is generated. In such a case, they are full also on their return trip. We assume that structural imbalances of import and export flows over time caused by containers moved between inland ports or through alternative deep sea ports can be ignored. For a commodity $c \in \mathcal{C}$, the desired service level is assumed to be expressed as a minimum weekly frequency $f^c$ of transport services offered.

To facilitate our modeling, we write:

$$
d_j^c = \begin{cases} d^c, & j = D^c \\ -d^c, & j = O^c \\ 0, & \text{otherwise} \end{cases} .
$$

**Table 4.** Commodity Parameters.

| | |
|---|---|
| $O^c$ | Origin of commodity $c$, $O^c \in \mathcal{N}$ |
| $D^c$ | Destination of commodity $c$, $D^c \in \mathcal{N}$ |
| $f^c$ | Minimum frequency of commodity $c \in \mathcal{C}$ |
| $d^c$ | Expected demand in TEU of commodity $c \in \mathcal{C}$ |

### 4.1.3. Cost and Capacity Parameters

We assume that cost of the uni-modal road transport alternative is linear in volume and denoted by $c_{ij}$ for all $(i,j) \in \mathcal{A}$. The container handling charges at transshipment nodes are also linear in volume and are denoted by $e_i$ for all $i \in \mathcal{N}$. The main difference between multi-modal and uni-modal road transport is that in the former, handling is performed twice (both at the seaport and the inland terminal) compared to only once in the latter (seaport handling only).

We consider a set of barges, $b \in \mathcal{B}$, with different cost and capacity characteristics. The cost of operating barges, from a barge operator's perspective, consists of several components, such as assets, crew, fuel, and maintenance [29]. On the other hand, the cost faced by a dry port operator that does not use its own barges is the price scheme proposed by the barge operator, which consists of the above cost components plus a markup. The leasing cost of a barge for a week is denoted by $W^b$ for all $b \in \mathcal{B}$ and includes both asset and staff cost required to navigate and operate the barges. Economies of scale apply in this leasing cost when higher capacity barges are selected; crew cost for barge navigation and operation are concave in the capacity of the barge. A variable cost $v_{ij}^b$ for all $(i,j) \in \mathcal{A}, b \in \mathcal{B}$, is also considered to represent the fuel cost of barges, which depends on the size (capacity) $Q^b$ of the barge $b$.

**Table 5.** Cost and Capacity Parameters.

| | |
|---|---|
| $Q^b$ | Capacity in TEUs of barge $b \in \mathcal{B}$ |
| $W^b$ | Weekly cost for leasing barge $b \in \mathcal{B}$ |
| $v_{ij}^b$ | Variable cost of barge $b \in \mathcal{B}$ traveling in arc $(i,j) \in \mathcal{A}$ |
| $e_i$ | Transhipment cost at node $i \in \mathcal{N}$ |
| $c_{ij}$ | Trucking cost per TEU for traveling link $(i,j) \in \mathcal{A}$ |

### 4.1.4. Time Parameters

We assume that the travel times $t_{ij}^b$ of barges on arcs $(i,j) \in \mathcal{A}$ may differ among barges $b \in \mathcal{B}$ and they may also include average delays due to congestion. We also consider a handling time $h_i$ per container loaded or unloaded on a barge in order to assess the minimum time spent on a call at a terminal which may depend on the terminal $i \in \mathcal{N}$. Finally, delays $l_i$ faced at seaport and inland terminals consist of mooring and un-mooring times, but also the actual delays faced at calls while waiting for sufficient space and time to berth.

**Table 6.** Time Parameters.

| | |
|---|---|
| $t_{ij}^b$ | Transportation time of barge $b \in \mathcal{B}$ traveling on arc $(i, j) \in \mathcal{A}$ |
| $l_i$ | Expected delay at terminal $i \in \mathcal{N}$ |
| $h_i$ | Handling time for loading/ unloading a TEU at terminal $i \in \mathcal{N}$ |

### 4.1.5. Decision Variables

We can separate the decision variables into four different sets according to their use. First, the Boolean variables $Y^b$ denote whether barge $b$ is selected and $y^{b,r}$ denote whether barge $b$ will perform route $r$. Second, we have variables associated with the construction of the routes. The Boolean variables $s_i^{b,r}$ denotes whether node $i$ is part of the route $r$ of barge $b$ and $m_{ij}^{b,r}$ whether arc $(i, j) \in \mathcal{A}$ will be part of the route $r$ of barge $b$; these two variables are connected via the vehicle routing constraints. Third, we have variables associated with the assignment of flows to specific transport services. The Boolean variables $g_c^{b,r}$ denote whether demand associated with commodity $c$ will be satisfied by barge $b$ in route $r$, while $z_{ij,c}^{b,r}$ represents the number of TEU of commodity $c$ on barge $b$ on route $r$ traveling on arc $(i, j)$. The number of TEU of commodity $c$ transported by trucks is denoted by $w_c$. Finally, we have continuous time variables $t_i^{b,r}$ that denote the arrival time of barge $b$ on route $r$ on node $i$, and $u^{b,r}$ that denote the end time of route $r$ of barge $b$ or, alternatively, the time the barge becomes available for its next route.

**Table 7.** Decision Variables.

| | |
|---|---|
| $Y^b \in \{0, 1\}$ | Denoting whether barge $b \in \mathcal{B}$ is used |
| $y^{b,r} \in \{0, 1\}$ | Denoting whether route $r \in \mathcal{R}$ will be performed by barge $b \in \mathcal{B}$ |
| $s_i^{b,r} \in \{0, 1\}$ | Denoting whether node $i \in \mathcal{N}$ will be called by barge $b \in \mathcal{B}$ on route $r \in \mathcal{R}$ |
| $m_{ij}^{b,r} \in \{0, 1\}$ | Denoting whether arc $(i, j) \in \mathcal{A}$ will be used by barge $b \in \mathcal{B}$ on route $r \in \mathcal{R}$ |
| $g_c^{b,r} \in \{0, 1\}$ | Denoting whether demand of commodity $c \in \mathcal{C}$ will be partly satisfied by barge $b \in \mathcal{B}$ in route $r \in \mathcal{R}$. |
| $z_{ij,c}^{b,r} \in \mathbf{Z}_{>0}$ | Amount of TEU of commodity $c \in \mathcal{C}$ transported in arc $(i, j) \in \mathcal{A}$ by barge $b \in \mathcal{B}$ on route $r \in \mathcal{R}$ |
| $w_c \in \mathbf{Z}_{>0}$ | Amount of TEU from commodity $c \in \mathcal{C}$ transported by trucks |
| $x_{i,c}^{b,r} \in \mathbf{Z}$ | Amount of TEUs transshipped in node $i \in \mathcal{N}$ for commodity $c \in \mathcal{C}$ and barge $b \in \mathcal{B}$ on route $r \in \mathcal{R}$ (loaded when positive, unloaded when negative) |
| $t_i^{b,r} \geq 0$ | Arrival time of barge $b \in \mathcal{B}$ in route $r \in \mathcal{R}$ at node $i \in \mathcal{N}$ |
| $u^{b,r} \geq 0$ | End time of route $r \in \mathcal{R}$ of barge $b \in \mathcal{B}$ |

### 4.2. MILP Formulation

In this section, the mixed integer linear program formulation is presented.

$$\min \sum_b Y^b W^b + \sum_b \sum_r \sum_{ij} m_{ij}^{b,r} v_{ij}^b + \sum_b \sum_r \sum_n \sum_c \left| x_{i,c}^{b,r} \right| e_i + \sum_c c_{O^c D^c} w_c \tag{1}$$

$$\sum_b \sum_r \sum_j z_{ij,c}^{b,r} - \sum_b \sum_r \sum_j z_{ji,c}^{b,r} = \begin{cases} d^c - w_c & i = O^c \\ 0 & else \\ w_c - d^c & i = D^c \end{cases} \quad \forall i \in \mathcal{I}, \forall c \in \mathcal{C} \tag{2}$$

$$x_{i,c}^{b,r} = \sum_j z_{ij,c}^{b,r} - \sum_j z_{ji,c}^{b,r} \quad \forall i \in \mathcal{I}, \forall c \in \mathcal{C}, \forall b \in \mathcal{B}, \forall r \in \mathcal{R} \tag{3}$$

$$\sum_j m_{ij}^{b,r} = s_i^{b,r} \quad \forall i \in \mathcal{I}, \forall b \in \mathcal{B}, \forall r \in \mathcal{R} \tag{4}$$

$$\sum_j m_{ji}^{b,r} = s_i^{b,r} \quad \forall i \in \mathcal{I}, \forall b \in \mathcal{B}, \forall r \in \mathcal{R} \tag{5}$$

$$\sum_c z_{ij,c}^{b,r} \leq Q^b m_{ij}^{b,r} \qquad \forall (i,j) \in \mathcal{A}, b \in \mathcal{B}, \forall r \in \mathcal{R} \tag{6}$$

$$\sum_r y^{b,r} \leq Y^b M \qquad \forall b \in \mathcal{B} \tag{7}$$

$$\sum_{ij} m_{ij}^{b,r} \leq y^{b,r} M \qquad \forall (i,j) \in \mathcal{A}, \forall b \in \mathcal{B}, \forall r \in \mathcal{R} \tag{8}$$

$$t_j^{b,r} - t_i^{b,r} \geq m_{ij}^{b,r} t_{ij}^b + \sum_i \sum_c h_i \cdot x_{i,c}^{b,r} + l_i - M \left(1 - m_{ij}^{b,r}\right) \qquad \forall (i,j) \in \mathcal{A}, \forall b \in \mathcal{B}, \forall r \in \mathcal{R} \tag{9}$$

$$u^{b,r} \geq t_i^{b,r} \qquad \forall i \in \mathcal{I}, \forall b \in \mathcal{B}, \forall r \in \mathcal{R} \tag{10}$$

$$t_i^{b,r+1} \geq u^{b,r} \qquad \forall i \in \mathcal{I}, \forall b \in \mathcal{B}, \forall r \in \mathcal{R} \tag{11}$$

$$2 \cdot g_c^{b,r} \leq s_{O^c}^{b,r} + s_{D^c}^{b,r} \qquad \forall c \in \mathcal{C}, \forall b \in \mathcal{B}, \forall r \in \mathcal{R} \tag{12}$$

$$f^c \cdot z_{ij,c}^{b,r} \leq d^c \cdot g_c^{b,r} \qquad \forall (i,j) \in \mathcal{A}, c \in \mathcal{C}, \forall b \in \mathcal{B}, \forall r \in \mathcal{R} \tag{13}$$

The objective function (1) minimizes all incurred costs. The costs consist of the weekly costs of leasing barges, the variable costs associated with the routing of barges, the handling costs for loading and unloading barges and finally the trucking costs for containers that are going to be transported by trucks. The objective function is bounded from above since all containers can be moved by trucks to their final destination. Constraints (2) stand for the flow conservation constraints while with constraints (3), the loading and unloading volumes of barges are calculated. Constraints (4) and (5) are the vehicle routing constraints that guarantee that an arc starts and ends at each node that is called in every round-trip. Constraints (6) are the capacity constraints for barges. Constraints (7) allow a round-trip to be constructed for a barge only if the barge is selected, while constraints (8) allow arcs to open only when a round-trip is opened.

The arrival times of barges at specific nodes are formulated in constraints (9) as follows; if barge $b$ on route $r$ travels on link $(i,j)$ the arrival time on node $j$, $t_j^{b,r}$, is equal to its arrival time on node $i$, $t_i^{b,r}$, enhanced by the travel time between nodes $i$ and $j$, the handling time of loading and unloading containers at node $i$ and the delay occurred at node $i$. Constraints (10) and (11) ensure time continuation between successive trips of barge $b$.

Constraints (12) ensure that a commodity can be assigned to a route of a specific barge only if both its origin and destination nodes are called on during the specific route. Constraints (13) represent the demand balancing or minimum frequency constraints. These constraints ensure that not all weekly demand can be consolidated in a single barge trip, which makes sense since demand arrivals are spread over the week so several services per OD pair are performed per week and consolidation is performed mainly with other commodities. The minimum frequency is used here also as a consolidation factor: The higher it is, the less consolidation can be achieved with a single commodity.

## 5. Case Study and Results

We develop an experiment to assess how the optimal fleet selection and routing decisions change under different parameter settings. We develop a realistic case based on a network service design problem of Brabant multi-modal (BIM), which was an alliance of five dry ports located in the Brabant region of The Netherlands in the proximity of the ports of Rotterdam and Antwerp. In particular, we analyze a part of the network in which BIM provides transport services, as depicted in Figure 2. The case of cooperation for capacity sharing for three dry ports is considered. These dry ports are OCT in Oosterhout, BTT in Tilburg and ITV in Veghel, which are connected to the port of Rotterdam through the same waterway. Barges can start from either BTT or ITV, and pass by the OCT terminal with a very small detour. Data as reported in this paper are realistic but present a modified and stylized version of the actual data, since the latter were confidential to BIM. The organisational aspects of this cooperation is beyond the scope of this paper, but we mention that one of the aspects that needed attention was the

redistribution of costs and benefits. Other cooperative arrangements between chain partners in the hinterland of the port of Rotterdam along corridors can be witnessed [34].

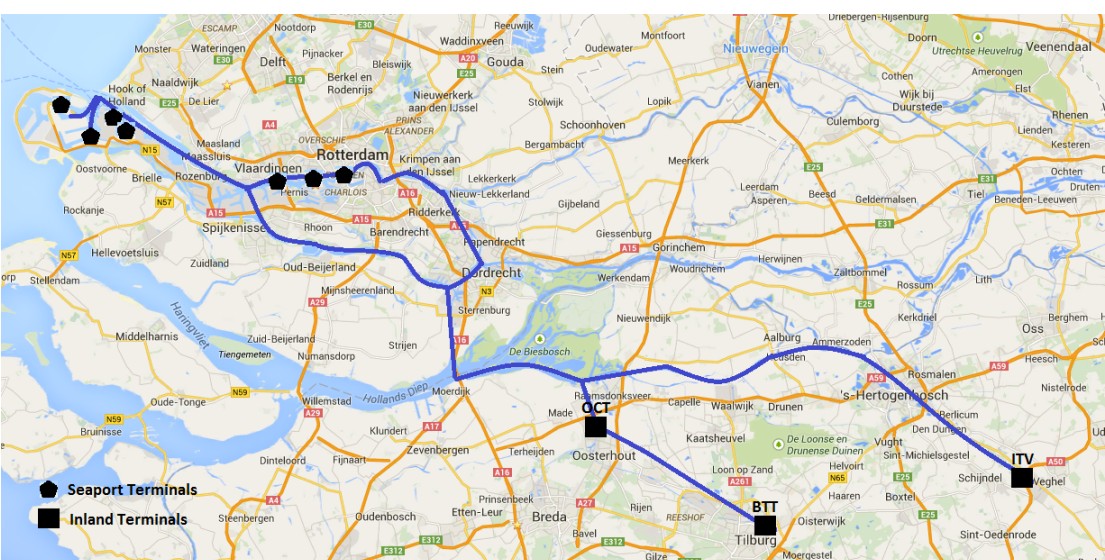

**Figure 2.** Map of seaport and inland terminals.

The experiment presented in this section is not meant to solve the actual problem of BIM, but to depict the capabilities of the model presented in the previous section, and moreover to identify design characteristics and assess how these characteristics affect the optimal design. We focus on four dimensions: (1) cooperation of inland terminals, (2) the consideration of service frequency constraints, (3) the demand volume, and (4) the expected delays before the barges berth at the seaport terminals. By analyzing the results, we show for each of the four dimensions how the optimal solutions change in terms of performance and fleet selection and routing decisions.

We apply the combination of the two models to the case at hand for illustration purposes. The models in combination would be applicable to any case where transport services are defined between two clusters of transport terminals, and where round trips (or vehicle routes) are performed within each cluster.

### 5.1. Experimental Set-Up

The data used in the experiments is provided in Appendix A. The demand considered for our case is presented in Table A1. Terminal parameters (delays, handling times and costs) are presented in Table A2. Barge fleet parameters (capacity, weekly leasing costs, sailing costs and times) are presented in Tables A3 and A4. Road transport parameters are presented in Table A5.

To assess the impact of cooperation we first apply our model to each dry port individually and aggregate the solution. We then apply our model for the case where all dry ports cooperate. We run 8 scenarios that correspond to different demand volumes (High-Low), delays at seaport terminals (Moderate-Severe) and Minimum frequency (0 or 4 times per week).

The different scenarios are coded with four characters codes, in the $a_1a_2a_3a_4$ format:

1. $a_1 \in \{I, C\}$: $C$ denotes the case of cooperation of inland terminals and $I$ denotes the aggregated solution of each inland terminal considered independently.
2. $a_2 \in \{M, S\}$: $M$ denotes scenarios with moderate delays at terminals and $S$ with severe delays.
3. $a_3 \in \{N, Y\}$: $N$ denotes scenarios with no minimum frequency constraints and $Y$ with minimum frequency constraints.
4. $a_4 \in \{H, L\}$: $H$ denotes scenarios with high demand and $L$ with low demand as denoted in Table A1.

## 5.2. Results

The mathematical model presented in Section 4 was formulated with the commercial IBM ILOG CPLEX 12 software. The instances in our experimental set-up were solved with the default branch and cut algorithm of the solver. We summarize the results of our experiments in Table 8. Some of the results are also graphically illustrated in Figures 3–9.

**Table 8.** Results MILP model.

| Scenario | Capacity Installed | Total Cost | Network Coverage | Modal Split Barge | Number of Barges | Average Barge Size | Average Calls/ Trip |
|---|---|---|---|---|---|---|---|
| IMNL | 810 | 31,207 | 1.00 | 100.0% | 3 | 90 | 2.33 |
| IMNH | 1500 | 36,968 | 1.50 | 100.0% | 3 | 110 | 2.29 |
| IMYL | 480 | 41,544 | 1.67 | 51.1% | 2 | 60 | 3.50 |
| IMYH | 1320 | 55,114 | 2.92 | 82.3% | 3 | 110 | 3.92 |
| CMNL | 750 | 16,235 | 1.00 | 96.1% | 1 | 150 | 3.20 |
| CMNH | 1500 | 29,245 | 1.25 | 99.2% | 2 | 150 | 2.50 |
| CMYL | 630 | 31,925 | 3.33 | 84.3% | 2 | 90 | 5.00 |
| CMYH | 1410 | 41,217 | 4.25 | 100.0% | 3 | 130 | 4.55 |
| ISNL | 810 | 31,207 | 1.00 | 100.0% | 3 | 90 | 2.33 |
| ISNH | 1500 | 38,760 | 1.25 | 100.0% | 3 | 150 | 2.50 |
| ISYL | 0 | 44,000 | 0.00 | 0.0% | 0 | 0 | 0.00 |
| ISYH | 1440 | 67,911 | 4.00 | 98.4% | 6 | 90 | 4.00 |
| CSNL | 660 | 23,189 | 1.25 | 100.0% | 2 | 120 | 3.17 |
| CSNH | 1350 | 28,964 | 1.33 | 98.4% | 2 | 150 | 2.78 |
| CSYL | 450 | 38,745 | 2.50 | 63.5% | 2 | 90 | 5.00 |
| CSYH | 1230 | 54,897 | 3.50 | 88.7% | 4 | 90 | 3.27 |

In Figure 3, the costs resulting from each scenario are presented. There is a clear cost benefit resulting from cooperation among the dry ports; the cost savings range form 12% to 48%. Moreover, as expected, minimum frequency constraints or higher delays increase overall costs. Not all drivers of cost increase have been clarified, so we need to further analyze the solutions.

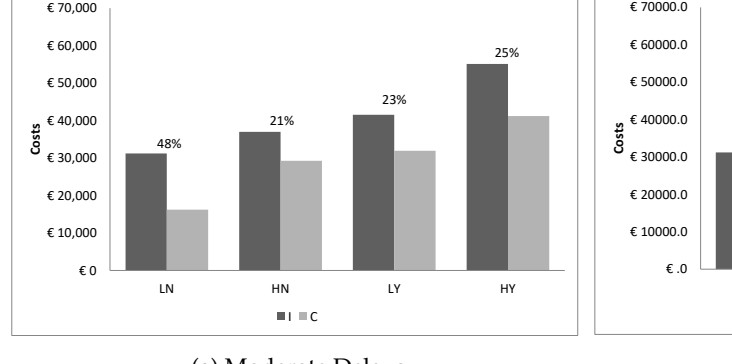

(**a**) Moderate Delays

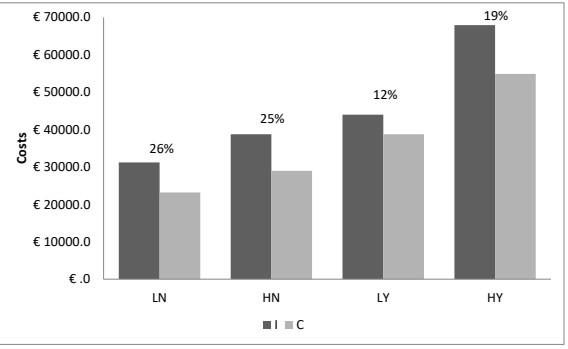

(**b**) Severe Delays

**Figure 3.** Cost benefits from dry port cooperation are indicated by comparing Independent (I) versus Cooperative (C) inland terminals. In both graphs, four results are compared where demand is either High (H) or Low (L), and minimum frequency constrains apply (Y) or not (N).

First, we look at the optimal fleet selection for each scenario in Figure 4. In case of cooperation, fewer but larger barges are selected in the optimal solutions compared to the case where each dry port is considered individually, so economies of scale are achieved. Imposing minimum frequency constraints also has an impact on the optimal fleet selection: smaller barges have to be deployed so that more round-trips can be achieved. This is particularly clear when high delays at the seaport terminals are considered. There are some cases where this is not realized though: For the scenarios with low demand and minimum frequency constraints, we observe that in the optimal solutions, only a small part of the demand is satisfied via barges, while the rest is satisfied via trucks, as seen in Figure 4. The relationship between network coverage and the percentage of demand satisfied by barges is shown in Figure 5, in which the consideration of minimum frequency constraints separates the optimal solutions in two distinctive groups. The scenario ISYL with Independent inland terminals, Severe delays, Minimum frequency requirements and Low demand, stands out, as all demand is satisfied by trucking. In Figure 5, one can observe that when the minimum frequency constraints are imposed, network coverage and modal split are related, which is not the case when these constraints are not applied.

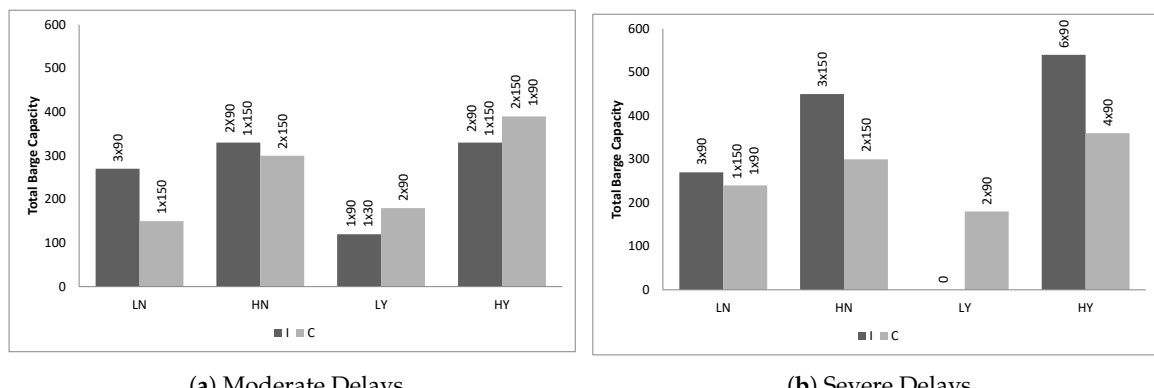

(**a**) Moderate Delays          (**b**) Severe Delays

**Figure 4.** Fleet Selection is compared between Independent (I) and Cooperative (C) inland terminals. In both graphs, four results are compared where demand is either High (H) or Low (L), and minimum frequency constrains apply (Y) or not (N).

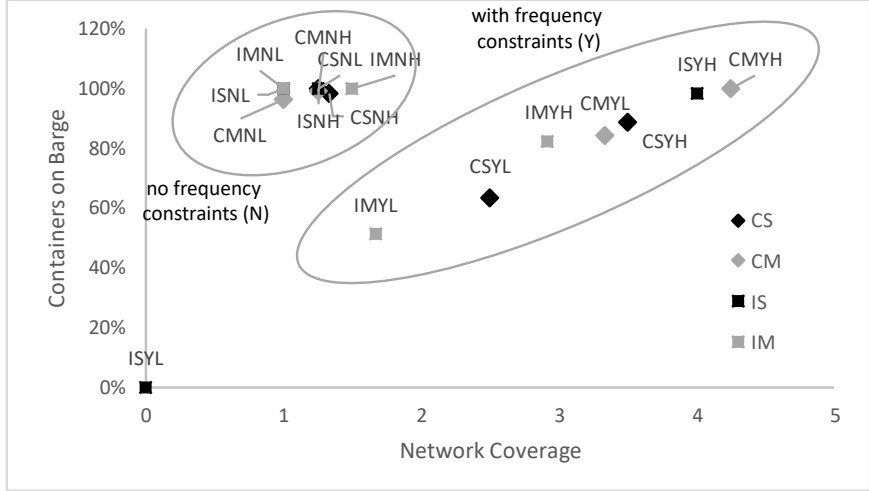

**Figure 5.** Network Coverage vs Containers on barge. Four types of scenarios are plotted where inland terminals cooperate (C) or work Independent (I), and where delays at terminals are Moderate (M) or Severe (S). Each type of scenario is plotted for minimum frequency constraint apply (Y) or not (N), and for Low (L) and High (H) demand.

Second, we consider for all scenarios the number of round-trips and number of calls at both seaport and inland terminals visited during each trip. This is shown in Figure 6. The total number of round-trips decreases and average barge size increases when cooperation is considered, except for the cases where a big share of demand is satisfied with trucking. Minimum frequency constraints also seem to have a great effect on the construction of round-trips. Round-trips are constructed with more calls at seaport terminals so that each round-trip satisfies smaller batches of demand of a greater number of OD pairs. This is more apparent in Figure 7, which contrasts the average number of terminals visited with the average network coverage. Here the consideration of minimum frequency constraints separates the solution in two distinctive groups. In Figure 6, it is shown that when cooperation is considered, there seems to be a clear advantage for barges to add calls at inland terminals, where delays are considerably shorter, as compared to adding calls at the congested seaport area.

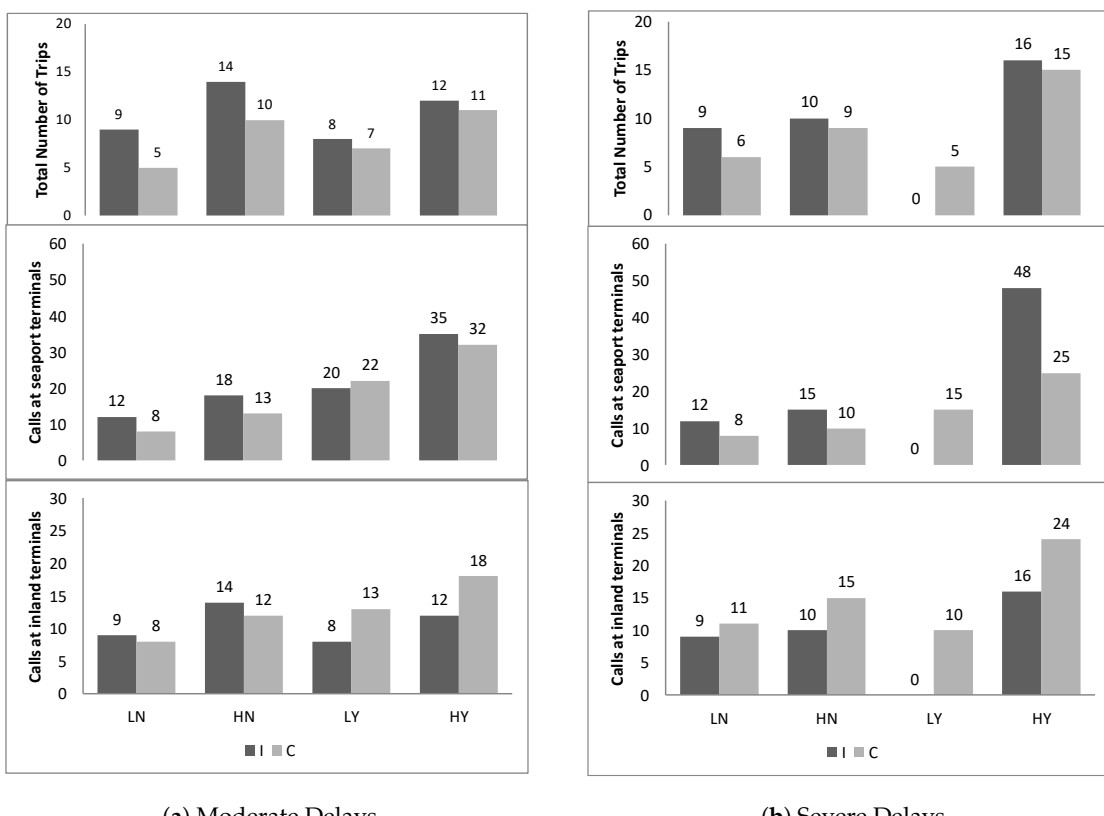

(**a**) Moderate Delays          (**b**) Severe Delays

**Figure 6.** Round-trips and calls at seaport and inland terminals are compared between Independent (I) and Cooperative (C) inland terminals. In both graphs, four results are compared where demand is either High (H) or Low (L), and minimum frequency constrains apply (Y) or not (N).

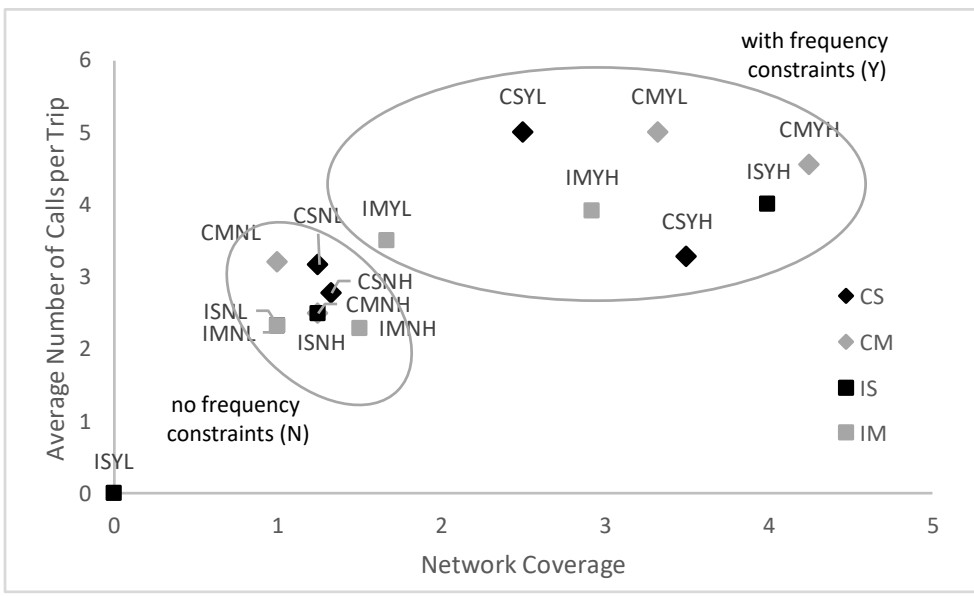

**Figure 7.** Network Coverage vs Number of Calls. Four types of scenarios are plotted where inland terminals cooperate (C) or work Independent (I), and where delays at terminals are Moderate (M) or Severe (S). Each type of scenario is plotted for minimum frequency constraint apply (Y) or not (N), and for Low (L) and High (H) demand.

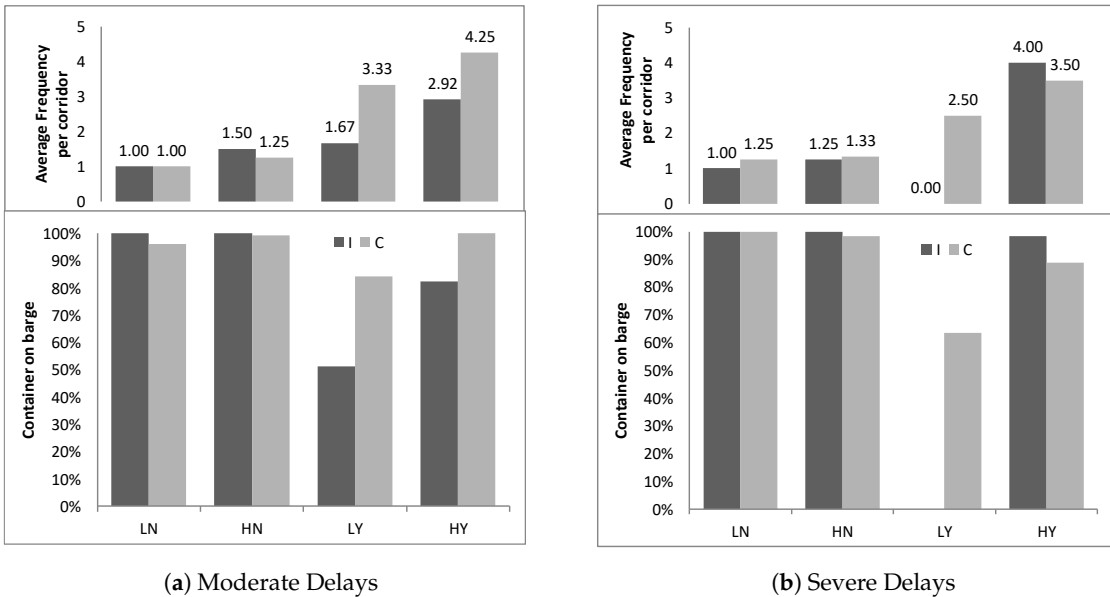

(**a**) Moderate Delays

(**b**) Severe Delays

**Figure 8.** Frequency on corridors and flow assignment is compared between Independent (I) and Cooperative (C) inland terminals. In both graphs, four results are compared where demand is either High (H) or Low (L), and minimum frequency constrains apply (Y) or not (N).

Last, we consider the quality of the service provided by each network configuration by looking at the average frequency of services provided for the OD pairs and the percentages of demand satisfied by barges and by trucks in Figures 5 and 8. We contrast the network coverage with the capacity installed in Figure 9, in which again the solutions are separated in two distinctive groups by minimum frequency constraints. When no minimum frequency constraints are considered, almost all demand is satisfied by barge trips, and the frequency of services for each OD pair increases only when it is

dictated by higher demand. In that way, barges call at fewer terminals and have lower circulation times, allowing for more round-trips. On the other hand, when minimum frequency constraints are imposed, smaller batches of demand for each OD pair must be consolidated for round-trips to be efficient. Of course, considering minimum frequency constraints makes solutions more realistic since it is hardly ever the case that weekly demand of an OD pair can be satisfied by one or two itineraries.

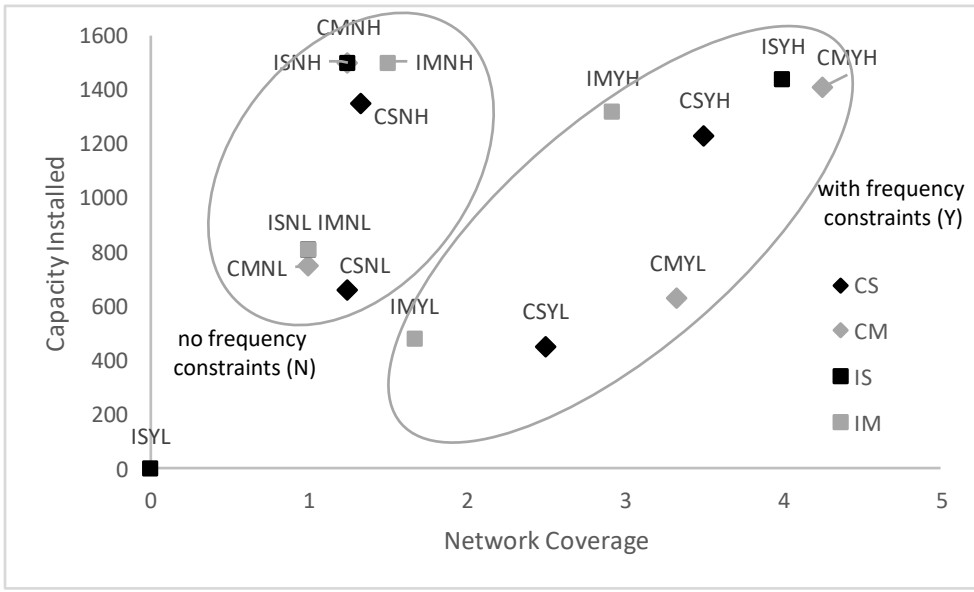

**Figure 9.** Network Coverage vs Capacity Installed. Four types of scenarios are plotted where inland terminals cooperate (C) or work Independent (I), and where delays at terminals are Moderate (M) or Severe (S). Each type of scenario is plotted for minimum frequency constraint apply (Y) or not (N), and for Low (L) and High (H) demand.

*5.3. Experimental Results Summary*

The effects of the main experimental dimensions on the optimal barge network design and performance are summarized in Table 9. In the table, an upward arrow ⇑ means that the respective parameter increases, a downward arrow ⇓ indicates that the parameter decreases, and the arrow ⇕ indicates that the parameter may either increase or decrease. In most cases, the stylized analytical model discussed in Section 6 shows the same effects, except for the two (downward) arrows with an asterisk, where the results of the stylized analytical model are mixed.

**Table 9.** Effect of Variables on Optimal Network Design and Performance: From (I)ndependent to (C)ooperative dry ports, from (N)o to (Y)es minimum frequency requirements, from (M)oderate to (S)evere delays, and from (L)ow to (H)igh demand.

| | Capacity Installed | Total Cost (per TEU) | Network Coverage | Modal Split Barge | Number of Barges | Average Barge Size | Average Calls/Trip |
|---|---|---|---|---|---|---|---|
| I → C | ⇕ | ⇓ | ⇕ | ⇕ | ⇓ | ⇑ | ⇑ |
| N → Y | ⇓ | ⇑ | ⇑ | ⇓ | ⇕ | ⇓* | ⇑ |
| M → S | ⇓ | ⇑ | ⇕ | ⇕ | ⇑ | ⇓* | ⇕ |
| L → H | ⇑ | ⇑ | ⇑ | ⇑ | ⇑ | ⇑ | ⇕ |

Our results show that all dimensions considered have a significant effect on the optimal design of barge service networks. The optimal network design can be considered to be case specific, and thus

only some conclusions can be drawn on how the different characteristics affect the optimal solutions. We develop an analytical model in the next section to study how these design trade-offs interact more specifically.

## 6. Stylized Analytical Model and Discussion

Optimal fleet selection and routing is always case specific and will depend on several characteristics such as expected demand, delays, distances, and available resources, as demonstrated by the MILP in Section 4. To better understand and appreciate the outcomes of the optimization model, we discuss in this section a simplified analytical model that demonstrates how the different design parameters affect the different performance characteristics. At the end of this section, we compare the outcomes of the analytical model with the outcomes of the optimization model. This will allow us to intuitively understand how the optimal solutions are driven by basic trade-offs already captured by the analytical model, and where the optimal solutions are tuned to more complex features of the decision problem at hand.

The analytical model presented in this section is not completely equivalent to the MILP model presented in Section 4, since several assumptions and simplifications were made: (a) demand is equally distributed among OD pairs, (b) the design variables (number of barges, barge size, number of calls, frequency) are assumed to be continuous, (c) the actual cost data are replaced by approximating continuous cost functions, (d) the circulation time of a round-trip is assumed to only depend on the number of calls and not the routing itself, (e) all demand can and will be served by barges.

### 6.1. Analytical Expressions

The model presented in this section provides some analytical expressions that connect performance indicators, such as cost, capacity and network coverage with different design parameters, such as size of barges, number of calls per round-trip, distribution of calls between seaport and inland terminals.

The notation used in this section and the values of some parameters fixed in this study are summarized in Table 10. A network is considered consisting of the node sets $\mathcal{N}^I$ and $\mathcal{N}^S$ denoting inland and seaport terminals respectively, with $N^I = |\mathcal{N}^I|$ and $N^S = |\mathcal{N}^S|$. It follows that $N^I \cdot N^S$ undirected $(O, D)$ pairs are considered, each associated with some demand that will be satisfied by several transport services. Vessels of different types are considered, resulting in an average barge size. The fixed (leasing) and variable (routing) costs of barges are assumed to depend on their size, which allows us to model economies of scale. The barges are assumed to perform round-trips continuously over the planning horizon. The round-trips are characterized by their expected average circulation time, $CT$, which is inversely proportional to the expected number of round-trips that a barge can perform during the planning horizon. The expected circulation time (CT) is calculated in (14) as a fixed sailing time, $\tau$, connecting the seaport with the hinterland areas enhanced by the variable delay times, $d_I$, $d_S$, associated with the additional time needed for calling at inland and seaport terminals (sailing, mooring, unmooring, handling, delays). In our case we consider a hinterland and a seaport area with terminals in each area located relatively close to each other. As a result, the main difference in the variable times is due to the higher delays faced at the seaport terminals. Considering the above, it is clear that the expected circulation time of round-trips is connected with the average number of inland terminals $n_I$, and seaport terminals, $n_S$. The number of calls and the distribution among inland and seaport terminals also affects the expected number of OD pairs that are served per round-trip, $SOD$, as calculated in Equation (16).

**Table 10.** Notation and Numerical Values Used.

| Sets and Costs | Decision Variables |
|---|---|
| $N^I$: Number of inland terminals (3 used) | $Q$: Average barge size |
| $N^S$: Number of Seaport Terminals (4 used) | $x$: Number of barges |
| | $n_r$: Average number of terminals visited per round-trip |
| $W$: Cost of leasing barge. $W = f_w(Q) \approx u_1 + u_2 Q$ | $n_I$ and $n_S$: Average number of inland and seaport |
| (Economies of scale) $W = 5000 + 50Q$ (used) | terminals visited per round-trip |
| $v$: Variable cost per round-trip of a barge. | $p_I$: Percentage of calls at inland terminals so that |
| $v = f_v(Q) \approx u_3 + u_4 Q$ (Economies of scale) | $n_I = n_r p_I$ and $N_S = n_r (1 - p_I)$ |
| $v = 300 + Q$ (used) | |
| **Time** | **Performance Indicators** |
| $T$: Planning horizon (168 h) | $SOD$: Average number of OD pairs served per round-trip |
| $\tau$: Fixed time per round-trip (16 h) | $CT$: Average circulation time of a round-trip |
| $d_I$, $d_S$: Variable time per call in a round-trip | $RT$: Average number of round-trips per barge |
| ($d_I$: 2 h, $d_s$: 4 or 8 h) | |
| | $TotalCost$: Estimated cost of plan |
| | $CapInst$: Port hinterland capacity installed over |
| | the network |
| | $NetworkCoverage$: Average service frequency per OD pair |

The three performance indicators, namely the expected Total Cost, Capacity Installed, and Network Coverage are calculated by means of formulas (17)–(19).

The expected total cost (17) is calculated as the product of the average fixed and variable cost of a barge per planning horizon and the number of barges. The fixed and variable costs are calculated with functions that feature economies of scale and depend on the average size of the barges used. The capacity installed (18) is calculated as the product of the average barge size, the number of barges, and the expected number of round-trips. The network coverage measure (19) depicts the expected frequency of services per OD pair and is calculated as the product of the average number of OD pairs served per round-trip (16), the expected number of round-trips per barge (15), and the number of barges divided by the number of OD pairs considered.

$$CT = \tau + 2d_I n_I + d_S n_S \tag{14}$$

$$RT = \frac{T}{CT} \tag{15}$$

$$SOD = n_I \cdot n_S \tag{16}$$

$$TotalCost = (W + vRT)\, x = \left( f_w(Q) + f_v(Q)\, \frac{T}{\tau + 2d_I n_I + d_S n_S} \right) x \tag{17}$$

$$CapInst = Q \cdot RT \cdot x = Q \cdot \frac{T}{\tau + 2d_I n_I + d_S n_S} \cdot x \tag{18}$$

$$NetworkCoverage = \frac{SOD}{N^I N^S} \cdot RT \cdot x = \frac{n_I n_S}{N^I N^S} \cdot \frac{T}{\tau + 2d_I n_I + d_S n_S} \cdot x \tag{19}$$

We evaluate formulas (17)–(19) for a range of input parameters in order to identify how the different network design characteristics interact and affect the performance indicators of such a network.

*6.2. Analytical Optimization Model*

Considering the above analytical expressions we could formulate a nonlinear constrained optimization problem with the same format as the MILP model presented in Section 4. In that sense, we would have the following nonlinear problem.

$$\min_{Q, x, n_s, n_I} TotalCost\,(Q, x, n_s, n_I) \tag{20}$$

subject to:

$$CapInst\,(Q, x, n_s, n_I) \geq d \tag{21}$$

where $d$ denotes the demand volume or the minimum capacity to be installed over the network, and

$$NetworkCoverage\,(x, n_s, n_I) \geq f \tag{22}$$

where $f$ denotes the minimum sailing frequency per time period $T$.

The objective function was given by (17), while the constraint functions where given by (18) and (19), respectively.

The above problem has a unique optimal solution. Since the objective function, the Capacity Installed, and the Network Coverage are all increasing in $x$ and $Q$, the optimal values $x^*$ and $Q^*$ can be found as a function of $n_I$ and $n_S$ by solving (21) and (22) as binding constraints, i.e., by replacing the inequality signs by equality signs in these formulas. The above yields the optimal number of barges

$$x^*\,(n_I, n_S) = f\,\frac{\tau + 2d_I n_I + d_S n_S}{T}\,\frac{N_I N_S}{n_I n_S} \tag{23}$$

and optimal average barge size

$$Q^*\,(n_I, n_S) = \frac{d\,(\tau + 2d_I n_I + d_S n_S)}{T x^*} \tag{24}$$

respectively. By substituting (23) in (24), we obtain:

$$Q^*\,(n_I, n_S) = \frac{d}{f}\,\frac{n_I n_S}{N_I N_S} \tag{25}$$

The analysis above implies that the minimum attainable cost is a function of $n_I$ and $n_S$:

$$TotalCost^*\,(n_I, n_S) =$$
$$\left(f_w\,(Q^*\,(n_I, n_S)) + f_v\,(Q^*\,(n_I, n_S))\,\tfrac{T}{\tau + 2d_I n_I + d_S n_S}\right) x^*\,(n_I, n_S) \tag{26}$$

The Capacity Installed is, by definition, equal to demand $d$. Network Coverage is equal to minimum frequency $f$ in case the global optimum is attained for $0 \leq n_I \leq N_I$ and $0 \leq n_S \leq N_S$. In such as case, the optimal $n_I$ and $n_S$ can be derived by solving the system of equations that results from the partial derivatives of the cost function with respect to $n_I$ and $n_S$. Although the above can lead to some complex analytical expressions depending on the assumed cost functions $f_w$ and $f_v$, it can be easily approximated by using mathematical programming languages. In some cases, optimal solutions are attained for boundary values, and then the Capacity Installed and Network Coverage may attain other values.

We solved the scenarios from the same experimental set-up as the one discussed in Section 5 with the MILP model and we present the results in Table 11.

By analyzing the results of the analytical model, the barge design trade-offs become clear. The number of barges and the number of calls per round-trip is determined such that the minimum service frequency is achieved. The resulting service frequency exceeds the minimum required level only when its "free"; e.g., due to the minimum number of barges (scenarios IMNL and IMNH) where cooperation is not considered and at least one barge should be assigned for each inland terminal. The minimum service frequency is achieved first by increasing the number of calls, and thus the average number of demand OD pairs served per round-trip, which has a small impact on cost, and then by increasing the number of barges, which has a higher impact on cost. After a routing plan has been determined, the optimal barge size is determined such that the resulting capacity meets the assumed demand. Of course, this simplistic sequential optimization would not hold in the real case where

design variables can only take discrete values. Overall, the results of the simplified model make clear and verify our observations based on the results of the MILP model. Table 9 illustrates how variables differ between scenarios, and is for a large part consistent with the findings of the MILP model (Table 8) and of the analytical model (Table 11).

**Table 11.** Results analytical model.

| Scenario | Capacity Installed | Total Cost | Network Coverage | Modal Split Barge | Number of Barges | Average Barge Size | Average Calls/Trip |
|----------|-------------------|-----------|------------------|-------------------|------------------|--------------------|--------------------|
| IMNL | 650 | 25,420 | 2.42 | 100.0% | 3.00 | 41.17 | 2.84 |
| IMNH | 1300 | 31,576 | 1.50 | 100.0% | 3.00 | 72.22 | 2.00 |
| IMYL | 650 | 27,375 | 4.00 | 100.0% | 3.00 | 54.2 | 5.00 |
| IMYH | 1300 | 36,150 | 4.00 | 100.0% | 3.00 | 108.3 | 5.00 |
| CMNL | 650 | 13,560 | 1.00 | 100.0% | 1.00 | 127.7 | 3.08 |
| CMNH | 1300 | 20,593 | 1.00 | 100.0% | 1.00 | 255.3 | 3.08 |
| CMYL | 650 | 19,811 | 4.00 | 100.0% | 1.67 | 108.3 | 6.00 |
| CMYH | 1300 | 29,482 | 4.00 | 100.0% | 1.70 | 210.2 | 5.88 |
| ISNL | 650 | 26,598 | 1.29 | 100.0% | 3.00 | 42.13 | 2.00 |
| ISNH | 1300 | 33,567 | 1.29 | 100.0% | 3.00 | 84.26 | 2.00 |
| ISYL | 650 | 37,652 | 4.00 | 100.0% | 4.33 | 54.2 | 5.00 |
| ISYH | 1300 | 49,919 | 4.00 | 100.0% | 4.50 | 98.5 | 4.64 |
| CSNL | 650 | 14,975 | 1.00 | 100.0% | 1.00 | 162.5 | 3.50 |
| CSNH | 1300 | 23,710 | 1.00 | 100.0% | 1.06 | 297.32 | 3.37 |
| CSYL | 650 | 26,749 | 4.00 | 100.0% | 2.36 | 105.1 | 5.88 |
| CSYH | 1300 | 38,621 | 4.00 | 100.0% | 2.79 | 148.7 | 4.74 |

## 7. Conclusions

In this paper, the heterogeneous fleet selection and barge routing problem in a port-hinterland network, which connects a set of closely located seaport container terminals with a set of closely located dry port terminals, was introduced. The analysis of the problem was made while using an MILP optimization model and a stylized analytical model.

We formulated the MILP model at a tactical level such that the demand over a planning horizon is met by several services distributed over that planning horizon. The use of the barges is considered not only in terms of space use of their capacity, but also in terms of time, by considering a continuous time formulation of the model. We take advantage of the special structure of the problem and provide a tight formulation that limits the number of variables and enables the efficient construction of round-trips such that commercial solvers can be used to find near optimal solutions in relatively low computation times.

Based on a real business case in the Netherlands, we study the design and performance of the fleet deployment and service network both with the MILP and the analytical model. First, we assess the impact of cooperation through capacity sharing between closely located dryport terminals. Second, we assess the main design trade-offs in the optimal barge network design of such a case. In our experiment, we vary several parameters, such as the demand volume, the expected delays at terminals, and minimum service frequency requirements.

The analysis of the results indicates that for our case, cooperation will always lead to cost reductions varying from 12% to 48%. Organizational aspects of cooperation, which may include redistribution of costs and benefits among partners, are beyond the scope of this paper. However,

the model results could be used as an input for such discussion as it allows the identification of benefits of certain coalitions of inland terminals.

Regarding the trade-offs made in service network design, it turns out that bigger barges were deployed when cooperation was considered, while each barge in a round trip served more OD pairs but with smaller batch sizes; this is how a high service frequency for every OD pair is achieved. Moreover, in case of cooperation, barges in most round trips seem to call more at the inland terminals instead of at the congested seaport terminals.

The main driver of cost is the number and size of barges used. Network coverage is mainly affected by the number of barges and their rotation over the network. With bigger barges, economies of scale can be achieved. To use these barges well, however, longer round trips with more calls at terminals are needed. This leads to longer circulation times and higher in-transit times for cargo. On the other hand, smaller barges can be used effectively for the formation of frequent shuttle services that satisfy demand for a single or a few OD pairs. Shuttle services or routes with few calls usually achieve lower circulation times and more round trips can be realized within a given planning horizon. Moreover, the number of calls during a round trip can affect the scheduling complexity as much as the reliability of transport times. This illustrates our two main trade-offs: (1) *Fleet composition*, where the use of big barges helps to reap economies of scale and of small barges to operate at a higher frequency; and (2) *Fleet routing*, where routes with many stops to consolidate demand and to provide high frequency of services are compared with routes with few stops to have short circulation times.

Although the case developed is rather small and results cannot be generalized easily, the main trade-offs in such a design emerge. Our results demonstrate some features that seem to be critical for the tactical port hinterland network design that models in this regime should incorporate. Our paper extends existing literature on port hinterland network design in this direction by proposing models on the tactical fleet selection and barge routing design incorporating these critical features.

The research performed has its limitations. We have developed two modeling approaches. The formulation of the inter-modal network design problem as an optimization problem helped to identify the benefits of cooperation and the optimal solutions for a range of scenarios. The analytical model, which roughly gave the same outcomes as the optimization model, helped to identify important trade-offs in the inter-modal network design. The optimization model could be used as a decision support tool for a dry port operator, only if the model would be validated externally. The analytical model is very stylized and may be improved to better match the optimization model outcomes.

The research could be extended in various ways. First of all, the environmental impact (e.g., emissions) of the network design could be taken into consideration explicitly. One could introduce a second objective or environmental target as a constraint. This would allow the inclusion of a policy maker and the study of the impact of various policy measures on optimal network designs. Second, we could relieve some of the assumptions made in the current model formulation. For example, structural imbalances between import and export volumes could be considered, together with the movement of empty containers.

**Author Contributions:** The first author wrote the manuscript that forms the basis for this paper as part of his PhD thesis. The second author has supervised the first author in writing his PhD thesis and has edited the manuscript for publication in this journal.

**Funding:** This research was funded by Dutch Institute for Advanced Logistics (Dinalog) under project Ultimate https://www.dinalog.nl/.

**Conflicts of Interest:** The authors declare no conflicts of interest.

## Appendix A. Case Data

**Table A1.** Average weekly demand in OD pairs for Low/High scenarios.

|      | DDE     | DDW     | EMX     | APM   |
|------|---------|---------|---------|-------|
| BTT  | 90/180  | 40/80   | 80/160  | 35/70 |
| OCT  | 70/140  | 60/120  | 40/80   | 40/80 |
| ITV  | 70/140  | 30/60   | 55/110  | 35/70 |

**Table A2.** Terminal parameters.

|                  | Inland | Sea |
|------------------|--------|-----|
| $l_i$ Low (h)    | 2      | 5   |
| $l_i$ High (h)   | 2      | 12  |
| $h_i$ (h)        | 5      | 25  |
| $E_i$ (euro)     | 20     | 200 |

Here $E_i$ is the total handling costs per barge at terminal $i$, irrespective of call size.

**Table A3.** Barge fleet parameters.

|                | Large      | Medium            | Small             |
|----------------|------------|-------------------|-------------------|
| $Q^b$ (TEU)    | 150        | 90                | 30                |
| $W^b$ (k-euro) | 10         | 8                 | 6                 |
| $v_{ij}^b$     | $v_{ij}$   | $\frac{4}{5}v_{ij}$ | $\frac{2}{5}v_{ij}$ |
| $t_{ij}^b$     | $t_{ij}$   | $t_{ij}$          | $t_{ij}$          |

**Table A4.** Time (hrs)/ cost (euro) parameters barge transport.

| $t_{ij}/v_{ij}$ | Art    | OCT   |
|-----------------|--------|-------|
| BBT             | 5/250  | 1/50  |
| OCT             | 5/220  | -     |
| ITV             | 6/280  | 2/90  |

Transport between seaport terminals assumed to take 1 h and 10 euro, while transport between sea terminals and artificial node takes 1 h and 20 euro.

**Table A5.** Road transport cost parameters (euro).

| $c_{ij}$ | port |
|----------|------|
| BBT      | 70   |
| OCT      | 60   |
| ITV      | 75   |

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
