# Peer review of "Collaborative Fleet Deployment and Routing for Sustainable Transport"

_sustainability, doi:10.3390/su11205666_

Round 1

Reviewer 1 Report

I understand that the paper is connected to PhD thesis, however the newest literature regarding intermodal transport as:

https://doi.org/10.3390/app9122558

https://doi.org/10.3390/su11082383

should be included, especially that the last quoted paper was published in 2015. All the more reason, additional literature should be analyzed if the Authors write that “the design of port-hinterland multi-modal transport systems has only recently grasped the attention of the academic world.”

Modes in the table 2 might rather be called as means of transport. Modes are rail transport, road transport and inland water transport in the case of this paper.

The demand volume expressed in TEU is problematic since it is rather economical equivalent unit – quantities of different kind iof ITU (e.g. containers) should be given instead. Authors are asked to make an adequate comment.

The blank parts on page 8 might be excluded. 

If there are several parameters under the summation sign in the model, then each of these parameters should appear under a separate sum sign (MILP formulation).

Appendix A is situated in the article quite unfortunately - it is mixed with references.

The reader might be confused: data were confidential to BIM, so does the Appendix A present real facilities data or not? Author are asked to comment it in the paper.

How did the Authors obtain the results? Any software? Any methodology? They probably did not calculate the model manually.

It is suggested to enlarge fonts in figs. 3-5, 7 or to adjust each chart to width of a page individually.

Authors claimed: “On the one hand, the optimal network design can be considered to be case specific, and thus only some conclusions can be drawn on how the different characteristics affect the optimal solutions. We develop an analytical model in the next section to study how these design trade-offs interact more specifically.” Does not that mean that obtained solutions might be treated as suboptimal?

Since several assumptions and simplifications were made, does not that mean the model is completely different? How about the verification and validation of the model?

The Authors are asked to establish equations for the final units, especially in the case of (23)-(26).

Number of barges must be integers and not expressed as decimal.

Limitation and further research should be given in conclusion part as well.

Author Response

We would like to thank reviewer 1 for the useful comments.

I understand that the paper is connected to PhD thesis, however the newest literature regarding intermodal                              transport             as:             https://doi.org/10.3390/app9122558             and https://doi.org/10.3390/su11082383 should be included, especially that the last quoted paper was published in 2015. All the more reason, additional literature should be analyzed if the Authors write that “the design of port-hinterland multi-modal transport systems has only recently grasped the attention of the academic world.”

Thank you for the relevant comment and suggestions. We have added a number of recent papers, which we considered the most relevant to our paper, to the literature review in order to address this comment.

Modes in the table 2 might rather be called as means of transport. Modes are rail transport, road transport and inland water transport in the case of this paper.

We have adapted the formulation: we have replaced “mode” by “means of transport”.

The demand volume expressed in TEU is problematic since it is rather economical equivalent unit – quantities of different kind of ITU (e.g. containers) should be given instead. Authors are asked to make an adequate comment.

Thank you for your comment. We use the unit of TEU as it stands for a standard unit of volume which is commonly used in the container transport industry. For instance, the capacity of a barge is usually expressed in terms of TEU, despite the fact that a barge also has a cargo weight capacity and that multiple types of container are used. We make a comment to justify our choice of unit: “Not all containers transported are equal to an integer multiple of the standard loading unit TEU, and the amount of cargo carried by a transport means may be constrained by weight instead of volume. Nonetheless, in this paper, as is often done in practice and academic literature, we express both demand volumes and capacities (of barges) in TEU.” (LL71-74)

The blank parts on page 8 might be excluded.

We have done so.

If there are several parameters under the summation sign in the model, then each of these parameters should appear under a separate sum sign (MILP formulation).

We have done so.

Appendix A is situated in the article quite unfortunately - it is mixed with references.

We have repaired this text editing flaw.

The reader might be confused: data were confidential to BIM, so does the Appendix A present real facilities data or not? Author are asked to comment it in the paper.

We have modified our explanation to clarify matters by stating that the original data as provided by BIM has been modified. The data (in the appendix) as presented is realistic, but has been stylized somewhat, which allows the data to be published without confidentiality issues. In the section on Case Study and Results we write: “Data as reported in this paper are realistic but present a modified and stylized version of the actual numbers since the latter were confidential to BIM.” (LL378-379)

How did the Authors obtain the results? Any software? Any methodology? They probably did not calculate the model manually.

We have specified this now better: “The mathematical model presented in Section 3 has been formulated with the commercial IBM ILOG CPLEX 12 software. The instances in our experimental set-up were solved with the default branch and cut algorithm of the solver.” (LL412-414)

It is suggested to enlarge fonts in figs. 3-5, 7 or to adjust each chart to width of a page individually.

We have enlarged the fonts in these figures. We have used concise notation in the figures and provided explanation in the captions.

Authors claimed: “On the one hand, the optimal network design can be considered to be case specific, and thus only some conclusions can be drawn on how the different characteristics affect the optimal solutions. We develop an analytical model in the next section to study how these design trade-offs interact more specifically.” Does not that mean that obtained solutions might be treated as suboptimal?

Thank you for this comment. We have improved our explanation of how the two models support our findings and how the results should be interpreted: “The aforementioned optimization model provides optimal barge fleet deployment and services on the network.

However, we aim to analyze the structure of these optimal solutions in terms of trade-offs, and we will do so by means of an analytical model in Section 5. The analytical model can be considered a stylized version of the optimization model introduced in Section 4. The analytical model, due to its simple structure, helps us to better understand the trade-offs made in network design. Although the analytical model provides outcomes similar to those of the optimization model qualitatively (see Table 9), the analytical model does not provide optimal solutions. The optimization model is of value in that respect.” (LL122-129)

Since several assumptions and simplifications were made, does not that mean the model is completely different? How about the verification and validation of the model?

The data used in the experimental design is realistic and derived from actual case data. As explained in response to the previous remark, we have developed two models that show similar behavior across the experimental design (see Tables 8, 9 and 11); this may serve as a verification of the models, based on mutual consistency. We have not scrutinized the model outcomes as viable deployments and routings of actual fleets, and we also do not make this claim. We state that: “The experiment presented in this section is not meant to solve the actual problem of BIM, but to depict the capabilities of the model presented in the previous section, and moreover to identify design characteristics and assess how these characteristics affect the optimal design.” (LL383-385)

The Authors are asked to establish equations for the final units, especially in the case of (23)- (26).

We have provided some clarification of how the formulas (23)-(26) have been obtained.

Number of barges must be integers and not expressed as decimal.

In the analytical model, the ‘number of barges’ is a real number and not necessarily integer. This contributes to the observation that the analytical model is highly stylized. The paper clarifies that nonetheless, the qualitative results as depicted in Table 11 are quite similar to the results in Table 8, which can be inferred from the fact that both tables are consistent with Table 9.

Limitation and further research should be given in conclusion part as well.

Thank you for the suggestion. We have included these parts in the conclusion section.

Reviewer 2 Report

The manuscript ID: sustainability-561516 entitled  Collaborative Fleet Deployment and Routing for

Sustainable Transport is well structured.

In order to improve this research here below the authors can find some observations.

First, the authors have to include a section dedicated to the academics and managerial implications.

Second, the authors have to highlights the contribution of these results compared to previous studies. The authors can include a specific section.

Third, the conclusions are weak. I suggest to review this section.

Finally, I suggest to a review of the English language.

Author Response

Thank you for your useful comments. The pdf attached includes all responses to the comments by the review team.

Reviewer 3 Report

Dear Authors,
The research on multimodal transport on the Collaborative Fleet Deployment and Routing for Sustainable Transport seems to me to be an interesting approach. The search for alternatives to road transport, or support for this type of transport, is important in some areas that make this possible.

Introduction
When you talk on line 26 about road transport emissions and climate change, you have to be careful. The latest European regulations on pollution (E6) applied to road transport give very low values for particulate emissions and pollution. This is indicated to you because you will later compare it with inland waterway transport, whose machinery can pollute quite a lot depending on the case.
On line 28, it states that transport by barge and rail is more sustainable than transport by road. It would be advisable that in addition to the bibliographical reference it includes, it should provide some percentages of the type of transport used in Europe, or at least in the area studied in the article.
Some comment could be made as to whether other boats (tourist, passengers, etc.) exist on the waterway used by barges or whether they can have access to the water channel without any restrictions.
In this introduction, in line 58 I could include a small comment on what types of barges exist, which although the load capacity is noted in the assumptions, could be included in this place. We all know what a lorry can load, but we are not clear about barges.
77. I encourage you to detail more forcefully the interest of the article, why you have chosen this research and what you hope to get out of it.
The supposed cooperation of the dry ports? Who defines it, are they associated according to a regulation? Does the government intervene in this grouping? I believe that it could define in some way how this cooperation has come about.
117. If you have previously defined it as a large barge, this part would be better understood.
Literature review.
I think the literature review is well structured and fairly comprehensive. It would be interesting to comment if there are other places where studies of this type have been carried out, or if any sustainability solution is being applied, I am referring mainly to real cases documented as examples of organization.

360-366, I think that the coding of the variables should be changed, so that it is easier to interpret the results.
1 (I, C); 2 (H,L); 3 (1,4); 4(H,L), I think that the options of point "4" High and Low should be called differently, so as not to confuse them with point 2.

380-383, figure 4 is supposed to be understood as "For the scenarios with low demand and minimum frequency constraints, we observe that in the optimal solutions, only a small part of the demand is satisfied via barges, while the rest is satisfied via trucks, as is shown".
in Figure 4". I believe that in Figure 4, this aspect is not appreciated.

434. If the model is not exactly like the MILP presented in section 3, you should comment on it in section 3, or give that information first if you think it is necessary.

In table 11, there are decimal numbers with (,) and others with (.), write them all down with (,)

On the conclusions, a good overview is made of what the research entails and the results obtained are noted down. I think it is necessary to structure this point a little better, because the reader expects to read in three or four direct ideas what has been found and what repercussions should have. I invite you to leave the first part as it is, and then describe in a more direct way what the most important findings have been.

In short:
In general, the work is well structured and easy to follow. I think it would gain a little more interest if I commented something more relative to the territory studied, also something about whether the model proposed would be applicable to other similar cases. I also think that it is interesting to comment on issues regarding collaboration between ports and the platforms commented on, since a large part of the results indicate that collaboration would be behind greater efficiency and sustainability of transport. How is this collaboration achieved if they are independent organisations, is it easy to make them collaborate? Does the government act in this respect?

Translated with www.DeepL.com/Translator

Author Response

(The authors gave the same response as above.)

Round 2

Reviewer 1 Report

The risk of Authors is that they admit the first Author wrote PhD thesis on the topic; the second author acted as supervisor. It would suggest that the first author worked on the paper himself. It is a matter of IP protection - please, verify that.

Some of crucial elaboration on the topic are still omitted. 

Using ”ITU” commonly in the container transport industry does not mean it as correct. Originally, the purposes of ITU were slightly different, however I understand this as commonly repeated substantive inaccuracy. I suggest, however, that Authors deepen their research. For example, it is good enough for capacity, but inaccurate to space design.

I regret, I cannot agree with answer for comment: ”Number of barges must be integers and not expressed as decimal.” Number of barges cannot be real number - its load capacity could be.

The conclusion are still weak. 

For future, it might be better to mark the changes made to the content of the article.

Author Response

Please find our responses in the attached pdf.
